# Medical Biology of Cancer-Associated Fibroblasts in Pancreatic Cancer

**DOI:** 10.3390/biology12081044

**Published:** 2023-07-25

**Authors:** Annah Morgan, Michelle Griffin, Lionel Kameni, Derrick C. Wan, Michael T. Longaker, Jeffrey A. Norton

**Affiliations:** 1Hagey Laboratory of Pediatric Regenerative Medicine, Stanford University School of Medicine, Stanford, CA 94305, USA; agmorgan@stanford.edu (A.M.); mgriff12@stanford.edu (M.G.); leokmen@stanford.edu (L.K.); dwan@stanford.edu (D.C.W.); longaker@stanford.edu (M.T.L.); 2Division of Plastic and Reconstructive Surgery, Department of Surgery, Stanford University School of Medicine, Stanford, CA 94305, USA; 3Division of General Surgery, Department of Surgery, Stanford University School of Medicine, Stanford, CA 94305, USA

**Keywords:** cancer-associated fibroblasts (CAFs), pancreatic ductal adenocarcinoma (PDAC), tumor microenvironment (TME), extracellular matrix (ECM), heterogeneity, metastasis, proliferation

## Abstract

**Simple Summary:**

Pancreatic cancer is a very deadly form of cancer with a low survival rate. One important characteristic of pancreatic cancer is the presence of a dense tissue called desmoplastic stroma, which creates an environment that promotes tumor growth. Within this environment, there are specialized cells called cancer-associated fibroblasts (CAFs) that play a crucial role. They are a diverse group of cells with different functions and surface markers. When activated, CAFs support the invasion, spread, and growth of cancer cells, as well as affect the immune system. Scientists have been studying how CAFs interact with cancer cells and immune cells to understand how they contribute to tumor growth and spread. However, there is still a lot we do not know about CAFs and their role in pancreatic cancer. Further research is needed to better understand CAFs and their impact on the development and progression of pancreatic cancer.

**Abstract:**

Pancreatic cancer is one of the deadliest forms of cancer with one of the lowest 5-year survival rates of all cancer types. A defining characteristic of pancreatic cancer is the existence of dense desmoplastic stroma that, when exposed to stimuli such as cytokines, growth factors, and chemokines, generate a tumor-promoting environment. Cancer-associated fibroblasts (CAFs) are activated during the progression of pancreatic cancer and are a crucial component of the tumor microenvironment (TME). CAFs are primarily pro-tumorigenic in their activated state and function as promoters of cancer invasion, proliferation, metastasis, and immune modulation. Aided by many signaling pathways, cytokines, and chemokines in the tumor microenvironment, CAFs can originate from many cell types including resident fibroblasts, mesenchymal stem cells, pancreatic stellate cells, adipocytes, epithelial cells, endothelial cells, and other cell types. CAFs are a highly heterogeneous cell type expressing a variety of surface markers and performing a wide range of tumor promoting and inhibiting functions. Single-cell transcriptomic analyses have revealed a high degree of specialization among CAFs. Some examples of CAF subpopulations include myofibrotic CAFs (myCAFs), which exhibit a matrix-producing contractile phenotype; inflammatory CAFs (iCAF) that are classified by their immunomodulating, secretory phenotype; and antigen-presenting CAFs (apCAFs), which have antigen-presenting capabilities and express Major Histocompatibility Complex II (MHC II). Over the last several years, various attempts have been undertaken to describe the mechanisms of CAF–tumor cell interaction, as well as CAF–immune cell interaction, that contribute to tumor proliferation, invasion, and metastasis. Although our understanding of CAF biology in cancer has steadily increased, the extent of CAFs heterogeneity and their role in the pathobiology of pancreatic cancer remains elusive. In this regard, it becomes increasingly evident that further research on CAFs in pancreatic cancer is necessary.

## 1. Introduction

Pancreatic cancer has one of the lowest survival rates of all cancer types, with its incidence rate nearly equaling its mortality rate [1]. While pancreatic cancer is not one of the top ten most common types of cancer, it sits at seventh in the ranking of cancer-related deaths in industrialized nations and is ranked the third most common cause of cancer-related deaths in the United States [2]. According to the National Institute of Health (NIH), pancreatic cancer accounted for 3.2% of all cancer cases in 2022, and yet was responsible for 8.2% of all cancer deaths that year [1]. The incidence rate of pancreatic cancer has historically been higher in developed countries; approximately 1.5% of people in the United States are diagnosed with pancreatic cancer at some point in their life—with the likelihood increasing with age [3]. The 5-year relative survival rate for pancreatic cancer has remained relatively steady over the last 3 decades at approximately 10% [1]. Survival rate for pancreatic cancer is dependent on the stage, or the degree of lymph node and liver metastases [1]. The 5-year relative survival rate of pancreatic cancer that is in the localized stage, or confined to the pancreas, is approximately 44%. When the cancer has spread to nearby lymph nodes, also known as the regional stage, the survival rate drops to approximately 15%. In the distant stage, in which the cancer has metastasized to distant parts of the body, the survival rate is only 3% [1].

Sadly, the symptoms of pancreatic cancer usually do not appear until the cancer has metastasized and reached an advanced stage. Due to this, most cases are not diagnosed or addressed until the cancer has progressed to a more aggressive and difficult-to-treat stage [4]. Screening and treatment methods have not improved significantly over the last several decades, making primary preventative methods particularly important for survival [5]. Pancreatic cancer typically presents as one of the two most common subtypes: Pancreatic Adenocarcinoma (PDAC), the most malignant and common subtype, which occurs within the exocrine pancreas and accounts for approximately 90–95% of pancreatic cancer cases, or Pancreatic Neuroendocrine Tumor (PanNET), occurring within the endocrine tissue of the pancreas and accounting for less than 5% of pancreatic cancer cases [2]. The definite causes of pancreatic cancer are still unknown, though many risk factors have been identified. These risk factors include smoking tobacco, over-consumption of alcohol, poor diet, chronic pancreatitis, diabetes mellitus, as well as many genetic conditions [3].

The genetic mutations that are commonly associated with pancreatic cancer are mutations of the Kirsten rat sarcoma viral oncogene homolog (*KRAS*), tumor protein 53 (*TP53*), mothers against decapentaplegic homolog 4 (*SMAD4*), also known as depleted in pancreatic cancer 4 (*DPC4*), and cyclin-dependent kinase inhibitor 2 (*CDKN2*) genes—none of which can be remedied through currently available treatment methods [4,6]. Overall, the treatment options currently available are limited. Pancreatic cancers are extraordinarily complex genomically, epigenetically, and metabolically—involving the activation of multiple pathways and substantial intercellular crosstalk within the tumor microenvironment [6]. This results in many challenges when attempting to devise medical treatments. New drug combinations and multifaceted treatment plans are attempted constantly; some have been shown to prolong survival, but currently there has been no treatment that has consistently demonstrated a response [6]. Surgical resection of a localized tumor is the usual treatment plan, but due to the fact that, in most cases, pancreatic cancer is not detected until it has metastasized, many patients are unresectable. The long-term survival in these cases is extremely poor [2].

## 2. CAF and Fibroblast Heterogeneity

In all types of cancer, the tumor microenvironment and its many components are heavily involved in the initiation, proliferation, and metastasis of the cancer [7,8]. The tumor microenvironment is not made up solely of tumor cells, but instead includes a variety of different cell types and structures, such as immune cells, endothelial cells, blood vessels, extracellular matrix, and cancer-associated fibroblasts (CAFs) [8,9]. CAFs are a highly heterogeneous type of fibroblast that play many roles, both activating and inhibiting, within the tumor microenvironment. Of all stromal cell types within the tumor microenvironment, CAFs are the most abundant. A higher concentration of CAFs within the tumor is typically associated with a poorer prognosis [8,10].

For decades, CAFs were believed to be a homogeneous population of cells within tumors, with the sole function of composing the extracellular matrix and giving the tumor its hard, fibrotic form [7]. However, recent studies have revealed that CAFs have a high degree of heterogeneity and plasticity, with various surface markers and functions that contribute to the progression or inhibition of tumor growth [11]. CAFs are activated within the stroma of the tumor to carry out an assortment of regulatory actions. These actions include the secretion of growth factors that contribute to tumorigenesis, the modification of the ECM to maintain tumor structure, the promotion of angiogenesis via the secretion of pro-angiogenic factors, and the suppression of the innate or treatment-enhanced anti-tumor immune response [9,10]. While most of the current evidence suggests that CAFs are tumor-promoting in nature, the results of many recent studies indicate that they also have functions that are tumor-suppressing [12].

The heterogeneity of CAFs can be seen in the roles each subtype plays in the promotion and inhibition of pancreatic cancer progression [13]. Through single-cell RNA sequencing of multiple tissue types, three unique clusters of CAFs with grossly contrasting functions have been identified—steady state-like (SSL), mechanoresponsive (MR), and immunomodulatory (IM) CAFs [14]. MR CAFs are activated in response to mechanical stimuli within the TME, IM CAFs perform immune-regulating functions, and SSL CAFs are in a steady state and when needed become either MR or IM CAFs [14]. Within these clusters exist many subtypes of CAFs. The most relevant in the context of pancreatic cancer are inflammatory CAFs, myofibrotic CAFs, and antigen-presenting CAFs [15,16].

The first two CAF subtypes identified in pancreatic cancer stroma were inflammatory CAFs (iCAFs) and myofibrotic CAFs (myCAFs) [15]. iCAFs have immunomodulatory functions and are characterized by their ability to promote metastasis, angiogenesis, and to produce various inflammatory cytokines and chemokines such as interleukin-6 (IL-6), interleukin-8 (IL-8), interleukin-11 (IL-11), C-X-C motif chemokine ligand 1 (CXCL1), C-X-C motif chemokine ligand 12 (CXCL12), and others (Figure 1) [7,16,17]. Tumor cells secrete paracrine factors such as IL-1, which activates NF-κB signaling and expression of leukemia inhibitory factor (LIF) in iCAFs [18]. In return, LIF secreted by iCAFs activates the STAT3 signaling pathway in cancer cells [18]. MyCAFs are identified according to their periglandular location and expression of high levels of αSMA, *COL1A1* and low levels of IL-6 (Figure 1) [16,17,19]. In PDAC, the desmoplastic stroma is primarily composed of myofibroblasts [20]. Shortly after the identification of these two subtypes, a third was discovered: antigen-presenting CAFs (apCAFs). These apCAFs are characterized by their antigen-presenting capabilities and expression of Major Histocompatibility Complex II (MHC II), as well as CD74, suggesting that they play a role in tumor immunity (Figure 1) [16,17,21,22,23].

Single-cell RNA sequencing (scRNA-seq) has revolutionized our understanding of cellular heterogeneity within complex tissues, including the identification and characterization of CAFs in PDAC. scRNA-seq enables the comprehensive profiling of gene expression at the single-cell level, allowing for the identification of distinct cell types and the exploration of transcriptional states [15,24,25]. Fibroblasts become a specific type of CAF via transcriptomic and epigenetic changes that result in increased gene expression [14]. The scRNAseq definition of CAFs in pancreatic cancer involves identifying unique gene expression signatures that allow us to distinguish them from other cell types within the TME [26,27,28]. These signatures typically include genes associated with fibroblast activation, such as *ACTA2* (which encodes αSMA), or genes associated with ECM remodeling, such as *COL1A1*, and the secretion of factors involved in tumor-stromal interactions [19,27,28]. We know that CAFs are a highly heterogeneous type of cells, so there is not one ‘scRNAseq definition’ that encompasses all CAFs. However, scRNA-seq analysis has provided us with a deeper understanding of the heterogeneity amongst CAFs through identifying differences between subsets and their specific functional properties [23].

Two subtypes of CAFs, myCAFs and iCAFS, have been thoroughly defined via scRNAseq analysis. Spatially, these two populations are found in distinct locations in relation to cancer cells; myCAFs are typically found near cancer cells, while iCAFs are usually located at the periphery of the tumor in desmoplastic regions [23,28]. MyCAFs, expressing elevated levels of αSMA, and iCAFs, expressing low levels of αSMA, express many of the same common fibroblast markers such as COL1A1, COL1A2, FAP, and VIM, along with some less common fibroblast markers, *PDPN* and *DCN* [23,27]. While myCAFs and iCAFs share many genes and markers, there are also many differences between the two populations. iCAFs express higher levels of IL-6, IL-8, CXCL1, CXCL2, CCL2, and CXCL12, along with the expression of complement factor D (CFD) and matrix proteins such as lamin A/C (LMNA) and dermatopontin (DPT) (Table 1) [23]. PDGFRα, a commonly used marker for all CAFs, was found to be upregulated in iCAFs when compared to myCAFs. The expression of hyaluron synthases (HAS1 and HAS2) and the gene AGTR1, which encodes angiotensin II receptor type 1, is also specific to iCAFs [23]. In contrast, markers specific to myCAFs include *TAGLN*, *MYL9*, *TPM1*, *TPM2*, *MMP11*, and *POSTN (*Table 1) [23]. The signaling pathways enriched by the genes expressed by these specific CAF subtypes were also identified. myCAFs were found to be implicated in smooth muscle contraction, ECM modification, collagen synthesis, and focal adhesion signaling pathways [23]. Many inflammatory pathways are upregulated in iCAFs including the complement pathway, IFNγ response, TNF/NF-κB, IL2/STAT5, and IL6/JAK/STAT3 signaling pathways [23,28].

The subpopulation of apCAFs is a more recent discovery and has not been analyzed as thoroughly as myCAFs and iCAFs. Through scRNAseq analysis, Elyada et al. discovered and classified apCAFs as a subcluster that was distinct from myCAFs and iCAFs [23]. The genes expressed by these apCAFs belonged to the MHC class II family; these genes are typically expressed by antigen-presenting immune cells (APCs) (Table 1) [23]. Some of these genes include histocompatibility 2, class II antigen A, alpha (*H2-Aa*), and beta 1 (*H2-Ab1*) [23]. apCAFs also expressed many of the classic fibroblast markers such as *COL1A1*, *COL1A2*, *DCN*, and *PDPN* at levels like that of iCAFs and myCAFs, thus confirming that they are true fibroblasts [23]. Antigen presentation, antigen processing, fatty acid metabolism, and mammalian target of rapamycin complex 1 (MTORC1) signaling are all uniquely upregulated in apCAFs [23].

The two subtypes of CAFs that are most prevalent in PDAC are myCAFs and iCAFs [19]. Pancreatic cancer cells stimulate both my CAFs and iCAFs. myCAFs are more common but iCAFs have been identified in all studies using surface markers and single-cell gene expression [14]. myCAFs generate the desmoplasia that encircles the tumor [10]. In some studies, myCAFs inhibit tumor progression, while in other studies, they facilitate it [10,19]. myCAFs inhibit tumor progression through engulfing the tumor in a dense collagen matrix that constricts and blocks cancer cells from progression and invasion [10]. myCAFs facilitate tumor progression through creating an active barrier to treatment with radiation, chemotherapy, and immunotherapy [10]. iCAFs inhibit the immune response to pancreatic tumors through secreting factors that inhibit immune recognition and anti-tumor defense [19]. It is crucial that the exact mechanisms of each type of CAF are determined to improve anti-cancer treatment.

## 3. Cellular Origins of Pancreatic CAFs

In recent years, the identification of distinct CAF subtypes has been made possible via single-cell transcriptomic analysis, but the exact origins of these different CAF phenotypes are still widely disputed [15]. It has been shown that the differing origins of tumor cells result in different malignancies and epithelial phenotypes, so it makes sense to hypothesize that the heterogeneity of CAFs can be attributed to their various cellular precursors [9,14]. In pancreatic tumors, the origin of CAFs has been traced to many different sources, the most prevalent being resident pancreatic fibroblasts, pancreatic stellate cells (PSCs), and tumor-infiltrating mesenchymal stem cells (MSCs) [15]. Other cell types may also be recruited to enrich the CAF population and promote the growth of fibrous connective tissue around tumor cells [15]. These other CAF sources may include bone marrow via circulation, adipocytes—the conversion of adipocytes to fibroblasts has been recently demonstrated by various groups, pericytes, fibrocytes, smooth muscle cells, mesothelial cells, and both epithelial cells and endothelial cells—via the epithelial-to-mesenchymal or endothelial-to-mesenchymal transition (Figure 2).

Within the primary tumor’s tissue of origin there are resident fibroblasts which are involved in maintaining homeostasis and become activated for a short period of time when the tissue has been wounded. In cancerous tissues, these fibroblasts are being activated constantly, resulting in desmoplasia [15]. We know from transcriptomic and lineage tracing studies that these resident fibroblasts are the source of a majority of the CAFs present in the tumor microenvironment. The main factor that promotes the activation and mobilization of these resident fibroblasts into CAFs is transforming growth-factor beta (TGF-β) [9]. TGF-β is secreted by both tumor and stromal cells and causes the resident fibroblasts to undergo myofibroblastic differentiation in processes such as wound healing and cancer progression [7].

Another source of CAFs is bone-marrow-derived mesenchymal stem cells (MSCs). MSCs are the source of up to 20% of all CAFs within the TME [9]. MSCs are multipotent cells that are able to differentiate into many different cell types, including fibroblasts. MSCs migrate to inflamed areas such as the tumor microenvironment [31]. Tumor cells secrete cytokines, growth factors, and chemokines such as TGF-β, CXCL12, PDGF, FGF, and IL-6 that trigger the MSCs to differentiate into CAFs [7,32]. Other cell types can also promote the differentiation of MSCs into CAFs. Immune cells such as T cells and macrophages secrete cytokines and chemokines that stimulate the differentiation of MSCs into CAFs [33]. Additionally, IL-1β and hepatoma-derived growth factor (HDGF) trigger MSCs to secrete tumor-promoting cytokines that support tumor progression [31].

For many years, pancreatic stellate cells (PSCs) have been accepted as a major cellular precursor of CAFs in PDAC [20,34]. PSCs are a type of resident pancreatic cells that retain lipid droplets with the capacity to store vitamin A [15]. PSCs are responsible for promoting fibrosis in chronic pancreatitis and produce ECM proteins within the TME [15,19,32]. PSCs can be activated by extracellular signals including TGF-β, tumor necrosis factor—alpha (TNF-α), acetaldehyde, ethanol, along with other cytokines and growth factors [35,36]. When activated, PSCs take on a myofibroblast-like phenotype and express αSMA [15,20]. Bachem et al. was able to isolate and compare αSMA+ PSCs from both human PDAC and chronic pancreatitis samples, and revealed that they were morphologically similar, expressing collagen, fibronectin, and desmin [20,34]. Activated PSCs secrete several factors that support angiogenesis and cancer cell proliferation, and also provide necessary metabolites for PDAC development [35,36]. Alanine generated by PSCs is utilized by PDAC cells in fatty acid biosynthesis [35,36].

Endothelial cells are also a potential source of CAFs. The endothelial-to mesenchymal transition (EndMT) is induced by factors secreted from surrounding cancer cells which stimulate the mesenchymal differentiation of endothelial cells into CAFs [9,14,37]. During the EndMT, endothelial cells aquire a mesenchymal-like phenotype as well as invasive and migratory properties [37]. Zeisberg et al. found that TGF-β1 could induce the conversion of endothelial cells into fibroblast-like cells that contribute to the CAF pool [38]. It has also been shown that TNF-α induces human endothelial cells to undergo EndMT [37,38]. Adjuto-Saccone et al. treated cultured endothelial cells with TNF-α and found that it resulted in a reduction of endothelial marker CD31, altered the expression of mesenchymal markers, and induced a morphological change in the cells from an epithelioid shape to an elongated, fibroblast-like appearance [37]. TNF-α downregulates the expression of endothelial receptor TIE1 and over-expression of TIE1 prevents TNF-α induced EndMT, suggesting that TNF-α, through TIE1, is able to regulate this process [37,38].

Another potential source of CAFs in pancreatic cancer is the pancreatic epithelium via the epithelium-to-mesenchymal transition [14,39]. The epithelial-to-mesenchymal transition (EMT) is a process in which epithelial cells undergo a variety of changes resulting in highly mobile, fibroblast-like mesenchymal cells [40]. Through the TGF-β-mediated epithelial-to-mesenchymal transition (EMT), epithelial cells are able to differentiate into functional CAFs expressing FSP-1 and αFAP [9]. Fibroblast-specific protein 1 (FSP-1) and alpha fibroblast activation protein (αFAP) are cell surface proteins expressed by activated fibroblasts [41]. FSP-1 has been implicated in promoting tumor progression in various forms of cancer, including pancreatic cancer [20,41]. FSP-1 also plays a role in modulating immune functions and can influence the functions of immune cells such as T cells, myeloid-derived suppressor cells (MDSCs), and macrophages [41]. High levels of FSP-1 is typically associated with poor prognosis and aggressive tumor metastasis [42]. αFAP contributes to cancer progression through several mechanisms, such as ECM remodeling which facilitates tumor invasion, angiogenesis, and supplies nutrients and oxygen to tumor cells [43]. αFAP enhances the bioavailability of hepatocyte growth factor (HGF) and TGF-β through cleaving their binding proteins [44]. αFAP also plays a role in suppressing the anti-tumor immune response within the TME through recruiting immunomodulatory cells such as regulatory T cells (Tregs) and MDSCs [43,45].

One study utilized lineage tracing in mice to determine the origin of pancreatic CAFs. It was found that the splanchnic mesenchyme—the layer that surrounds the endoderm from which the pancreatic epithelium originates—is the major source of resident fibroblasts in the pancreas. These resident fibroblasts give rise to the majority of CAFs found in pancreatic tumors [13]. The lineage tracing experiment utilized a combination of the *Isl1cre* allele and a cre-dependent *R26Tomato* allele to label the cells of the splanchnic mesenchyme. After the development of PDAC, it was found that pancreatic CAFs also expressed Tomato, indicating that they are descendants of the splanchnic mesenchyme. Upon single-cell RNA analysis, it was revealed that specific gene expression signatures were shared among the splanchnic mesenchyme, resident fibroblasts, and CAFs [13]. The single-cell RNA sequencing analysis compared the gene expression of resident pancreatic fibroblasts, CAFs, and splanchnic mesenchyme from the E9.5 foregut and revealed a group of genes that were highly expressed in both CAFs and the splanchnic mesenchyme and another group of genes were highly expressed in both resident fibroblasts and the splanchnic mesenchyme [13]. The group of genes shared by resident fibroblasts may be involved in morphogenesis and homeostasis and the groups of genes shared by CAFs and the splanchnic mesenchyme may be implicated in fetal programs that are reactivated during tumorigenesis [13].

It was recently discovered that apCAFs are derived from mesothelial cells [21]. In the development of pancreatic cancer, mesothelial cells—derived from the embryonic mesoderm—lose their mesothelial features and gain fibroblast-like features through a process similar to the epithelial-to-mesenchymal transition (EMT) [21]. This process is driven by Nuclear Factor-κB (NF-κB) signaling, induced by IL-1, or transforming growth factor-β (TGF-β) signaling via the SMAD1, SMAD3, SMAD4 pathways [21].

Adipocyte-to-fibroblast transition has been studied and observed by many groups [20,46,47]. Mature adipocytes can undergo a process of de-differentiation and transdifferentiation to acquire a fibroblast-like phenotype [47]. In tumors, adipocytes are recruited by TGF-β and CXCL12 and then activated to transition into CAFs by TGF-β and PDGF through mechanisms like those that activate resident fibroblasts [7,47]. Kidd et al. observed the generation of adipocyte-derived CAFs and found that αSMA+ tumor stroma was generated from local adipose tissue [48]. In this study, mice received subcutaneous transplants of GFP+ adipose tissue. After 10 days post-transplantation E0771 murine cancer cells were injected adjacent to the transplanted adipose tissue. GFP+ cells were found throughout nearby tumor tissue but not in distant tumor tissue, indicating that those cells originated from the transplanted adipocytes [48]. The GFP+ cells were quantified, and it was found that a majority of the transplanted adipose tissue contributed to α-SMA, nerve/glial antigen 2 (NG2), and CD31 endothelial cell populations. The remaining adipose derived GFP+ cells were found to be αSMA+/NG2+ CAFs [48]. Griffin et al. demonstrated that adipocytes convert to fibroblasts in response to Piezo-mediated mechano-sensing, revealing that mechanical stimuli alone are enough to induce an adipocyte-to-fibroblast transition [46].
Figure 2Cellular Origins of CAFs in Pancreatic Cancer. This figure depicts the different cell types that can be activated by various growth factors, cytokines, and/or physiological conditions to transition into CAFs. These cell types include resident fibroblasts, pancreatic stellate cells, mesenchymal stem cells, epithelial cells, endothelial cells, adipocytes, pericytes, fibrocytes, and smooth muscle cells. These diverse cell sources contribute to the pool of CAFs in the TME. Some key markers associated with CAFs are alpha-smooth muscle actin (αSMA) and fibroblast specific protein-1 (FSP-1), also known as S100A4. This figure was reprinted with permission from Ref. [49]. 2019, Liu et al.
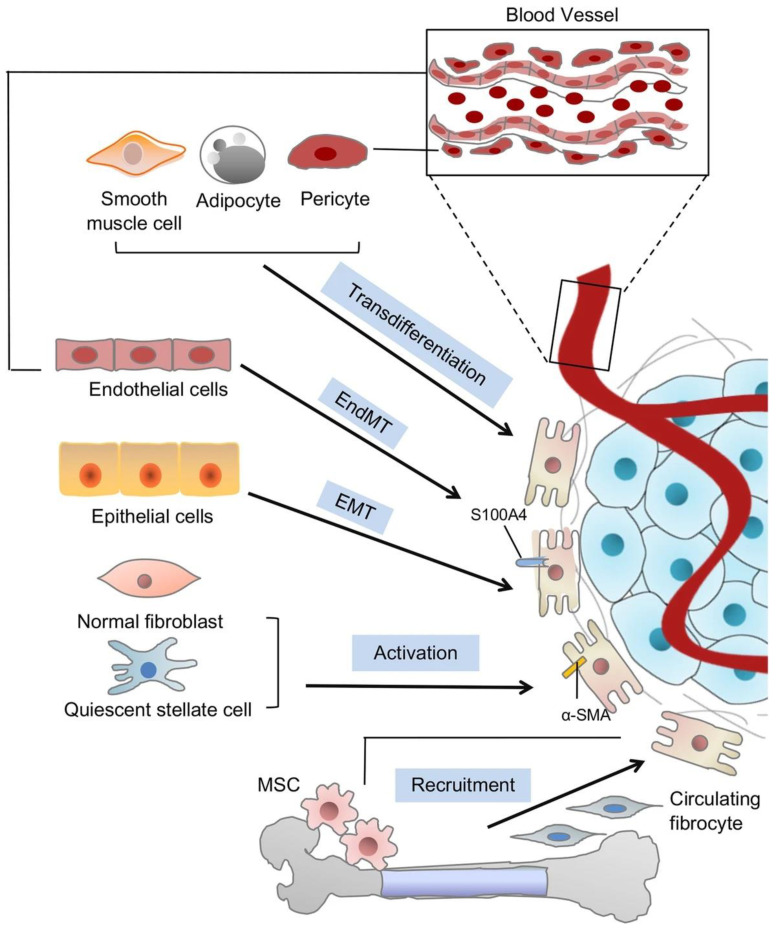



## 4. Cell Surface Markers of CAFs

In pancreatic cancer, CAFs express a number of cell surface markers that allow us to identify and distinguish them from other cell types in the TME. These markers can also help us understand their functionality. CAFs can be traced to many different cellular origins and there are many markers associated with each of them. These markers include, but are not limited to, Col1, αSMA, PDGFR, Vimentin, FSP-1, and FAP [7,9,10,31,50]. However, these markers are not unique to CAFs, or even fibroblasts, and are expressed by many other cell types [51]. There is a large degree of heterogeneity amongst CAF subpopulations and none of these individual markers are unique to any particular subset [10]. CAFs can also be characterized by their production of cytokines such as IL-6 and IL-8, as well as vascular endothelial growth factor (VEGF), a potent angiogenic factor [9].

Collagen type 1 (Col1) is a well-known marker commonly associated with CAFs. Col1 is a major component of the ECM and CAFs play a critical role in both establishing and remodeling the ECM of tumors [20,52]. The upregulation of CAFs is typically accompanied by an upregulation of Col1. Collagen provides structure to the TME and facilitates tumor cell movement. CAFs produce excessive amount of Col1 in the tumor stroma, resulting in the fibrotic and desmoplastic characteristics of pancreatic cancer [20,31]. Elevated levels of Col1 in pancreatic cancer are associated with increased tumor size, metastasis, and poor clinical outcomes [20].

Alpha smooth muscle actin (αSMA) is a distinct marker of CAFs and fibroblasts and is commonly used to identify them in pancreatic cancer. αSMA is involved in the contractile functioning of fibroblasts [10,20]. αSMA is a marker of myCAFs but is also expressed by many other CAF subtypes and in other cells like smooth muscle cells and pericytes [10]. Immunostaining studies have shown that there is significant correlation between αSMA and Collagen deposition in pancreatic cancer specimens [10,20].

In pancreatic cancer, Platelet Derived Growth Factor Receptor alpha (PDGFRα) and beta (PDGFRβ) are both expressed widely by fibroblasts and CAFs [53,54]. These receptor tyrosine kinases are involved in cell migration, proliferation, and tissue remodeling [20]. Elevated expression of PDGFRβ, specifically, is associated with chemoresistance, poorer prognosis, and higher tumor recurrence rates [53,54]. PDGFRα was found to be upregulated in iCAFs, specifically [23].

Vimentin (VIM) is a protein expressed by mesenchymal cells, such as fibroblasts, that plays a role in providing mechanical support, maintaining cellular integrity, cell signaling and migration [7,54]. CAFs are associated with an upregulation of VIM [54]. This distinguishes them from normal, inactivated fibroblasts in the TME [31]. VIM is implicated in many CAF functions, including ECM remodeling, metastasis, and immune modulation, along with enhanced contractility and secretion of tumor-promoting factors [54].

Fibroblast activation protein (FAP) is a cell surface glycoprotein that is considered a key marker of fibroblasts and CAFs. FAP is crucial in the modulation of the TME and is associated with tumor progression and ECM remodeling [43,55]. Over-expression of FAP is typically associated with a poor prognosis [43,45]. FAP is not exclusively a marker of CAFs and can be expressed by all activated fibroblasts, as well as by a subset of CD45+ immune cells [44]. When identifying CAFs with FAP, identification is typically confirmed using negative epithelial markers such as EPCAM [54].

CAFs also express the calcium-binding protein Fibroblast-Specific Protein 1 (FSP-1), also known as S100A4. FSP-1 is involved in cell migration and is a reliable marker for all fibroblasts, not just CAFs [20,52]. FSP1 is commonly used to direct inactive, non-proliferating fibroblasts [41]. It also identifies macrophages and is expressed by some pancreatic cancer cells [56].

When attempting to identify CAFs using these markers, it is necessary to consider the morphology and spatial distribution that is exhibited [52,56]. As mentioned, these markers are not exclusive to CAFs, or fibroblasts as a whole. They may not be expressed at the same time, and their expression can vary between different CAF subtypes [10]. CAFs are a diverse cell type with each subtype having distinct functional and phenotypic characteristics. Each subset of CAFs exhibit different patterns of marker expression depending on their cellular origins, activation states, and specific functions within the TME [54].

## 5. CAF Signaling Pathways

CAFs are responsible for much of the cellular communication that occurs within the TME [15,57]. These signaling pathways communicate and interact with each other, resulting in complex networks that control fibroblast activation and function. Any disruption in the regulation of these pathways can contribute to fibrosis, tumor progression, and metastasis [57].

A main function of CAFs in PDAC is to promote tumor growth and metastasis [58]. Factors secreted by CAFs activate signaling pathways such as the transforming growth factor—beta (TGF-β) pathway, platelet-derived growth factor (PDGF) pathway, Wnt/β-catenin pathway, fibroblast growth factor (FGF) pathway, Hedgehog pathway, and mitogen-activated protein kinase (MAPK) pathway in cancer cells that promote proliferation, survival, and migration [32,57,59].

The activation of resident fibroblasts and other precursor cells into CAFs is dependent on growth factors and signaling pathways. The TGF-β pathway is a major regulator of fibroblast activation [60,61]. TGF-β is one of the most prominent factors within the TME and is the main factor responsible for the activation of resident fibroblasts into CAFs [23,60]. TGF-β binds to receptors that activate downstream signaling pathways, such as the SMAD-dependent and SMAD-independent pathways, and leads to the transcriptional regulation of genes involved in fibroblast activation as well as ECM deposition and remodeling [22,52].

The PDGF and FGF pathways also involve growth factors that regulate fibroblast proliferation, migration, and differentiation [62,63]. Both of these pathways involve the binding of PDGF or FGF to receptors and result in activation of signaling cascades, like the RAS/MAPK pathways that lead to fibroblast activation and production of ECM proteins [62,63].

The Wnt/β-catenin pathway is implicated in regulating fibroblast activation and proliferation. This pathway involves 19 Wnt ligands and over 15 receptors, classified as either canonical or noncanonical signaling pathways [57,64,65]. Activation of this pathway results in translocation of β-catenin into the nuclei, where it regulates gene expression involved in fibroblast activation and ECM remodeling [57,64]. The Wnt/β-catenin pathway induces the differentiation and activation of myofibroblasts [65]. It interacts with other signaling pathways, such as the TGF-β and Hedgehog pathways, that modulate fibroblast activation and fibrogenesis [56]. The Hedgehog pathway also regulates fibroblast activation and is elevated in myCAFs [66].

The mitogen-activated protein kinase (MAPK) pathway is involved in many different cellular processes such as cell proliferation, differentiation, and migration [67]. The RAS/MAPK pathway, specifically, plays a significant role in pancreatic cancer. MAPK is involved in CAF activation, secretion of growth factors and cytokines, ECM remodeling, and overall tumor progression [68]. The MAPK pathway consists of several kinase cascades that are activated by different combinations of extracellular stimuli, growth factors, cytokines, and environmental stress [69]. Activation of CAFs through the MAPK pathway is triggered by factors such as TGF-β, PDGF, and FGF [70]. MAPK signaling regulates interactions between CAFs and immune cells within the TME and can also regulate the secretion of immune cell recruiting chemokines and cytokines by CAFs [70]. The MAPK pathway also contributes to therapeutic resistance in tumors through triggering CAFs to secrete factors that promote tumor cell survival and negate the cytotoxic effects of certain treatments [63,64,65].

## 6. Immune Suppression of CAFs

CAFs generally promote an immunosuppressive TME. Immunosuppression and drug resistance are significant tumor-promoting functions of CAFs (Figure 3). A cluster of CXCL12-expressing immunomodulatory CAFs was identified through scRNAseq analysis [14]. This cluster regulates a wide variety of inflammatory mechanisms within the TME [14]. Pathway analysis of immunomodulatory (IM) CAFs reveals an upregulation of the complement activation cascade, cytokine-mediated signaling pathway, and regulation of acute inflammatory response [14].

IM CAFs secrete a number of cytokines and growth factors that modulate cancer response to therapy [22]. IM CAFs create an immunosuppressive environment within the tumor through secreting CXCL12, TGF-β, GAS6, IL-6, HGF, GDF15, and FGF5, each of which promote cancer cell invasion, proliferation, and immunosuppression [22]. IM CAFs suppress the recruitment, polarization, and function of immune cells including monocytes, myeloid cells, CD8+ T cells, Treg cells, and macrophages through secreting IL-1β, IL-6, IL-8, CXCL9, CXCL12, CCL2, and TGF-β [22,71].

When co-cultured with CAFs, tumor associated macrophages transition from an anti-tumorigenic (M1) to a pro-tumorigenic (M2) functionality via activation by IL-4, IL-10, and IL-13 [72,73]. CAFs prevent CD8+ T cell infiltration into the tumor and recruit immunosuppressive cell types like myeloid-derived suppressor cells (MDSCs) and neutrophils [22,74]. CAFs can inhibit the function of NK cells, Th1 lymphocytes, and DCs via the secretion of TGF-β, CXCL1, and IL-10 cytokines [7]. CAFs are also able to suppress immune cells through direct contact due to their expression of βig-h3, a TGF-β-induced surface protein that binds to integrin β3 on the surface of CD8+ T lymphocytes, which inhibits their cytotoxic function [7].

The subpopulation of antigen-presenting CAFs (apCAFs) have the ability to induce naive CD4^+^ T cells into regulatory T cells (Tregs) through antigen-dependent T cell receptor ligation [21]. Tregs suppress the innate anti-tumor immune responses and promote tumor progression. Higher levels of Tregs in the TME are positively correlated with poorer outcomes [75]. It was found that these induced Tregs significantly inhibited the proliferation of tumor-suppressing CD8^+^ T cells [21].
Figure 3Tumor-promoting and tumor-suppressing effects of CAFs. This figure depicts the multiple cancer-promoting and cancer-suppressing behaviors of CAFs in pancreatic tumors, as well as the factors and markers associated with these functions. The tumor-promoting effects include chemoresistance, immunosuppression, cancer cell proliferation, angiogenesis, invasion, and ECM remodeling. The tumor-suppressing effects include chemosensitivity, decreased tumor malignancy, and decreased self-renewal of cancer cells. This figure was reprinted with permission from Ref. [76]. 2021, Mhaidly et al.
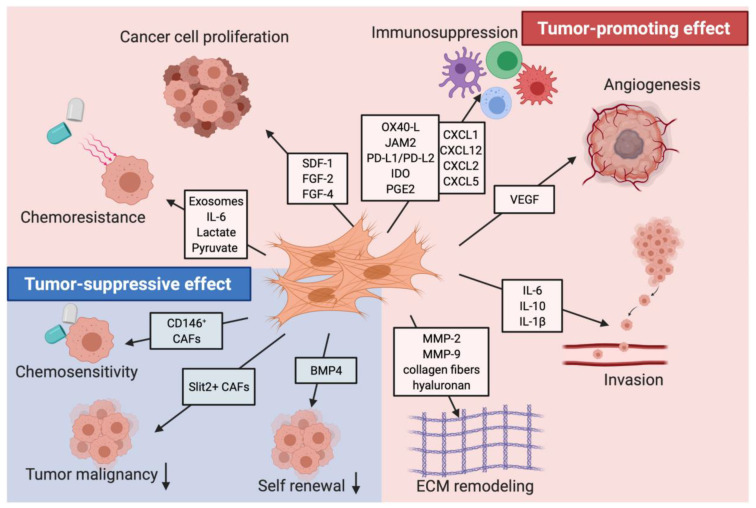



## 7. CAFs as Potential Anti-PDAC Treatment

Previous attempts to therapeutically deplete or inhibit proliferation of CAFs in PDAC were unsuccessful in both mice and humans [77]. Often, treatments targeting CAFs result in cancer progression rather than depletion. This suggests that some CAFs likely have tumor-suppressive functions while others have tumor-promoting functions [76,77].

While the functions of CAFs have been shown to be mostly tumor-promoting, they can also inhibit tumor progression (Figure 3.). CAFs may inhibit tumor growth through promoting an anti-tumor immune response. Multiple studies have shown that CAFs produce immunomodulatory cytokines such as IL-10, TGF-β, TNF, INF-gamma, IL-6, CXCL9, CXCL10, and CXCL11 that can also recruit immune cells such as T cells, macrophages, and natural killer cells to the TME [52]. CAFs can also stimulate the differentiation of monocytes into M1 macrophages that have anti-tumor properties [78]. In mouse models of PDAC, an αSMA expressing population of CAFs exhibited an increase of Shh signaling that inhibited production of VEGF, CXCL12, and IL-8—factors that promote angiogenesis, tumor growth, and immunosuppression in the TME [7,20]. In other mice models, targeting FAP+ CAFs resulted in the recruitment of CD8+ T cells and suppression of collagen synthesis, allowing immune control over tumor growth and an overall anti-tumor effect [79,80].

One subpopulation of CAFs has been classified as mechanoresponsive (MR) CAFs [14,81]. Mechanoresponsive CAFs exhibit heightened responsiveness to mechanical stimulation within the TME. They have been shown to express elevated levels of mechanosensitive signaling mediators, mechanosensors such as integrins and focal adhesion proteins, and express genes such as *Mgp*, *Gas6*, *Postn*, and *Fosb* that are associated with mechanotransduction and the focal adhesion kinase (FAK) signaling pathway [14]. Mechanoresponsive CAFs show an upregulation of focal adhesion, integrin binding, cell-matrix adhesion signaling pathways [14]. These MR CAFs actively participate in ECM remodeling and promote ECM stiffening via increased deposition of collagen and fibronectin fibers, that result in enhanced tumor stiffness and alterations of the tumor’s biomechanical properties [29]. Alterations of the ECM affect tumor behavior and promote tumor progression [29,30]. In response to mechanical cues, MR CAFs activate multiple signaling pathways involving TGF-β, Yes Associated Protein (YAP), and MAPK [29,30].

Mechanoresponsive CAFs are typically associated with poor clinical outcomes in pancreatic cancer. The presence of these CAFs is associated with increased tumor aggressiveness, therapeutic resistance, and decreased patient survival. MR CAFs are able to facilitate tumor cell invasion and metastasis through their ECM remodeling functions [29,30]. Many groups are now targeting mechanoresponsive CAFs and/or the signaling pathways involved in mechanotransduction, as these targets may offer new therapeutic strategies [14,29,30].

Many of the current therapeutic strategies used to target CAFs in pancreatic cancer aim to deplete certain CAF populations or reverse CAFs to their inactivated states, but some recent experimental treatment methods target the downstream effectors of CAFs [76,82,83]. For example, the inhibition of JAK2/STAT3 and MEK/ERK/1/2 by ruxolitinib and trametinib in a mouse model resulted in an anti-tumor response and increased overall survival [84]. Targeting CAF-derived cytokines and chemokines, such as IL-6 and TGF-β, may be effective in improving the innate anti-tumor immune response [82]. In immunotherapy treatments that co-administer a TGF-β inhibitor with anti-PD-L1, TGF-β signaling is reduced in stromal cells which facilitates T cell infiltration into the tumor and enhances the anti-tumor immune response [83].

Trials targeting CAFs, in general, have not been successful. However, some very recent studies have shown real progress in the identification of mechanisms through which CAFs induce resistance to both radiation and immunotherapy [85,86]. In general, CAFs have been identified to be a potential anti-tumor target through the demonstration of enhanced single-cell gene expression within a cancerous tumor [81]. Standard anti-cancer therapy targets tumor cells without consideration of potential damage to the tumor microenvironment (TME). In rectal cancer of both mice and humans, the presence of iCAFs is associated with a poor response to chemotherapy. In rectal cancer, interleukin-1 (IL-1) is a potential anti-tumor target because it predisposes iCAFs to senescence that causes chemoradiation therapy resistance and tumor progression. IL-1 inhibition with the IL-1 receptor antagonist (anakinra) prevents iCAF senescence and improves the response to radiation (in experimental rectal cancer) [85,86]. This experimental finding in mice has resulted in a prospective trial in patients with rectal cancer and could possibly be utilized in the treatment of other types of cancer, including PDAC [86].

With minimal responses to immunotherapy, a better understanding of the TME and CAF function is needed. Using scRNA transcriptomics, researchers have identified a population of CAFs that are programmed by TGF-beta to express leucine-rich repeat containing protein 15 (LRRC15) [87]. This protein was present in the circulation of 22 patients with PDAC, and elevated levels correlated with a poor response to anti-PD-L1 therapy [87].

The above two studies demonstrate how CAF function can negatively affect tumor responses to different treatments. In the former, CAF production of IL-1 inhibits response to chemoradiation therapy which is a major upfront treatment for patients with T-3 rectal cancer, along with many other cancer types [85,86]. In the second study we see how myCAF expression of LRRC15 negatively affects anti-tumor response with anti-PD-L1 immunotherapy [87]. Studies to inhibit this protein are underway [88]. Better understanding and discovery of CAF effects on therapy is necessary to properly manipulate CAF function to enhance antitumor effects.

## 8. Conclusions

Recent research on CAFs has gained a lot of attention and it has been suggested that targeting CAFs may be a promising novel strategy in the treatment of pancreatic cancer. We know that a higher concentration of CAFs in the TME is typically associated with a poorer outcome, so targeting CAFs instead of cancer cells could potentially enhance the response to certain drug treatments or the innate immune system. Drugs that target and inhibit the activity of CAFs or disrupt the stroma may improve the delivery of chemotherapy or immunotherapy to cancer cells and enhance the effectiveness of treatment. However, more research is needed to fully understand the role of CAFs in pancreatic cancer and develop effective therapies that target these cells.

CAFs are difficult to target due to all the varying subtypes that exist within the TME. Treatments targeting cells that CAFs originate from, the cytokines and chemokines that they express, or the signaling pathways involved in activating and carrying out their various functions can potentially be effective treatment strategies. Targeting these mechanisms could limit the differentiation of certain cells into CAFs and inhibit the tumor-promoting functions of CAFs. Further investigation into the origins of activated fibroblasts and the mechanisms behind their behavior is needed.

In conclusion, CAFs are a critical component of the tumor microenvironment in pancreatic cancer. They promote tumor growth, metastasis, drug resistance, and contribute to the evasion of the innate immune anti-tumor response. Recent advances in our understanding of the heterogeneity of CAFs in pancreatic cancer may lead to the development of new targeted therapies. However, much more research is needed to fully understand the complex interactions between CAF subpopulations and cancer cells in pancreatic cancer. Understanding the heterogeneity of CAFs and the role that each subtype plays in both cancer progression and inhibition is essential for the design and development of new targeted therapies and improved clinical outcomes for pancreatic cancer.

## Figures and Tables

**Figure 1 biology-12-01044-f001:**
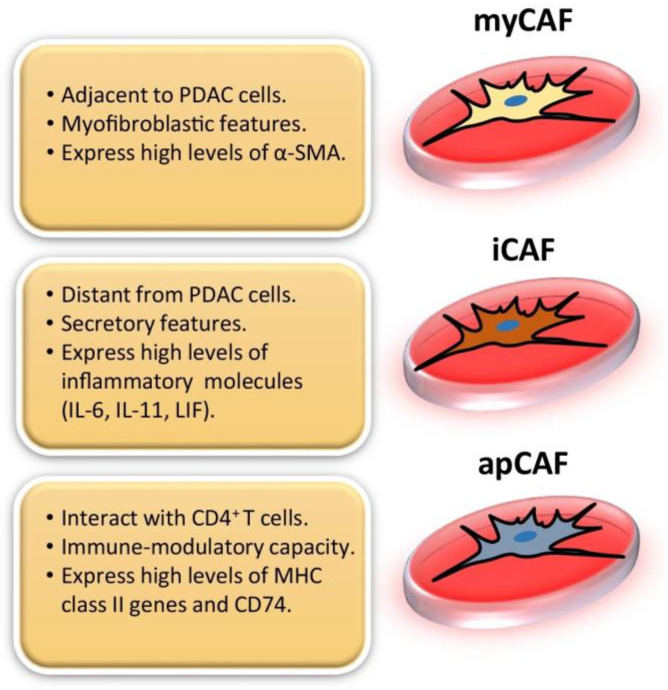
In pancreatic ductal adenocarcinoma (PDAC), there are three distinguishable subtypes of CAFs that have been identified and characterized: myofibroblastic CAFs (myCAFs), inflammatory CAFs (iCAFs), and antigen-presenting CAFs (apCAFs). This figure was reprinted with permission from Ref. [16]. 2022, Barrera et al.

**Table 1 biology-12-01044-t001:** Main markers and functions of relevant CAF subtypes in pancreatic cancer. This table summarizes the main markers and functions of three different CAF subtypes that are relevant in pancreatic cancer: myofibrotic CAFs (myCAFs), inflammatory CAFs (iCAFs), and antigen-presenting CAFs (apCAFs).

CAF Subtypes	Markers	Functions	References
myCAFs	αSMA, Col1a1, TAGLN, MYL9, TPM1, TPM2, MMP11, POSTN	ECM remodeling, Tumor invasion, proliferation, and metastasis	[10,14,23,29,30]
iCAFs	IL-6, IL-8, CXCL1, CXCL2, CXCL12, PDGFRα, CFD, CCL2,	Immune suppression, Chemoresistance	[11,14,15,17,23]
apCAFs	MHC II, CD74	Antigen presentation, Immune modulation	[15,17,22,23]

αSMA; alpha Smooth Muscle Actin, CCL2; C-C motif chemokine ligand 2, CD74; Cluster of Differentiation 74, CFD; Complement Factor D, Col1a1; Collagen type 1 alpha chain 1, CXCL1; C-X-C motif chemokine ligand, CXCL2; C-X-C motif chemokine ligand 2, CXCL12; C-X-C motif chemokine ligand 12, IL-6; Interleukin-6, IL-8; Interleukin-8, MHC II; Major histocompatibility complex II, MMP11; Matrix metalloproteinase 11, MYL9; Myosin light chain 9, PDGFRα; Platelet-derived growth factor receptor alpha, POSTN; Periostin, TAGLN; Transgelin, TPM1; Tropomyosin 1, TPM2; Tropomyosin 2.

## Data Availability

Not applicable.

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
