# Peer review of "Medical Biology of Cancer-Associated Fibroblasts in Pancreatic Cancer"

_biology, 2023, doi:10.3390/biology12081044_

Round 1

Reviewer 1 Report

This is a review of current knowledge of cancer-associated fibroblasts in pancreatic cancer. Please see my comments below.

Abstract

1.    The cells cannot express a variety of phenotypes or functions. They can express different surface markers. Please rewrite the following statement ”CAFs are a highly heterogeneous cell type expressing a variety of phenotypes, functions - both activating and inhibiting, and surface markers.”

2.    Please add immune cells in the following statement “Over the last several years; various attempts have been undertaken to describe the mechanisms of CAF-tumor cell interaction that contribute to tumor proliferation, invasion, and metastasis.”

General Statement

1.    Do not capitalize pancreatic cancer in the middle of the sentences.

2.    All citations should be listed before full stops, not after.

Introduction

1.    Lane 67, the second-word pancreas should be pancreatic

2.    Mutations refer to genes, and gene abbreviations should be italicized. In addition, please provide full names before employing abbreviations.

Section 2

1.    Lane 89, the abbreviation CAFs should be introduced in the sentence before when cancer-associated fibroblasts are mentioned for the first time.

2.    Lanes 98/99, The following statement is vague, and no reference is listed: “It has been suggested that CAFs originate from multiple tissue and cell types.”

3.    Lane 121, It would be interesting to provide more details on what the ligands secrete by pancreatic cancer cells and what pathway they activate to stimulate iCAFs in the following statement: Tumor cells secrete paracrine factors that activate iCAFs.

4.    Lane 123, Col1 should be modified to COL1A1 or Collagen.

5.    Lane 126, The following statement repeats to some extent information provided in the sentence before and after it. “As the name implies, these apCAFs have antigen-presenting capabilities.”

6.    How do these subtypes modulate pancreatic cancer progression and immune cell activity in PDAC? What is their role in stemness?

7.    Do the pancreatic cancer cells stimulate specific CAFs?

8.    What changes do cells undergo to become a specific CAF: transcriptomic, metabolomic, or epigenetic?

Section 3

1.    Lane 169, Please unify the nomenclature for TNF-α.

2.    Lanes 188-190. The authors stated, "Through the TGF-β-mediated epithelial-to-mesenchymal transition (EMT), epithelial cells are able to differentiate into functional CAFs expressing FSP-1 and αFAP” What are FSP-1 and AFP? What is their role? It is crucial to provide more context. This is true for the entire Section 3.

3.    Lane 194, how deeper single-cell RNA-seq differs from single-cell RNA-seq? What are these signatures obtained that are shared by splanchnic mesenchyme, resident fibroblasts, and CAFs?

4.    Lane 207, What were the conditions under which this transition was observed? "Kidd et al. observed the generation of adipocyte-derived CAFs." Please add more information so the Reader can understand the study and the outcome.

Section 5

1.    This section should be included in section 2 when the authors characterized different CAFs. Some of the information provided in Section 5 is a repetition of Section 2. Please move to incorporate section 5 into section 2.

Section 6.

1.    The entire first paragraph of section 6 is a repetition of other sections. At this stage, all Readers will know that CAFs are heterogenous and play a role in tumorigenesis.

2.    Lane 331, The authors stated, "Drugs targeting EGFR proteins are now being used in treatment of pancreatic cancer and other cancer types.” What are those drugs? What is the outcome of the treatment?

3.    WNT/catenin pathway is involved in cell proliferation.

4.    Lane 435, Can you provide examples?

How have CAFs been targeted? What was the outcome? Why did the trials fail? What are the current approaches to target CAFs? How do researchers study CAFs to develop new therapeutics?

Figures 1 and 2 and Table 1 are not cited in the manuscript.

This review manuscript has only 51 citations; eighty percent come from reviews, not original articles. Primary (original research) articles should be the bedrock of the review, with minimal citations from previous reviews. The manuscript needs more depth of the current knowledge of CAFs biology.

Please see comments included in the main critique.

Author Response

Thank you for your comments. We feel that they helped us to create a stronger, more in depth review. 

Reviewer 2 Report

Morgan et al., in the above article has described in detail the biology of CAFs and their importance or role in Pancreatic cancer. The review aptly provides sufficient information with regards to the different types of CAFs, their source of origin, different cell specific markers and the pathways they influence. More importantly, the review does point to the dichotomy in CAF function, one as a tumor promoter and other as a suppressor. However, I would ask the authors to include a small figure/schematic early on the review to show the different subtypes of CAF under the broad umbrella of CAFs (Kind of a hierarchical figure). Given so many different subtypes based on their source of origin and function, having a small schematic will definitely help the readers.

Author Response

(The authors gave the same response as above.)

Round 2

Reviewer 1 Report

The authors satisfactorily addressed my comments.

Author Response

Thank you for your comments, they have helped us to create a more comprehensive review.